# Membrane-Coated Biomimetic Nanoparticles: A State-of-the-Art Multifunctional Weapon for Tumor Immunotherapy

**DOI:** 10.3390/membranes12080738

**Published:** 2022-07-28

**Authors:** Yuanyuan Zhang, Xinyi Zhang, Haitao Li, Jianyong Liu, Wei Wei, Jie Gao

**Affiliations:** 1Changhai Clinical Research Unit, Shanghai Changhai Hospital, Naval Medical University, Shanghai 200433, China; zyy09150915@163.com; 2Institute of Translational Medicine, Shanghai University, Shanghai 200444, China; zhangxy_913@shu.edu.cn; 3Department of Vascular Surgery, Wuhan Union Hospital, Tongji Medical College, Huazhong University of Science and Technology, 1277 Jiefangdadao Road, Wuhan 430022, China; castiel9286@163.com (H.L.); liujianyonguh@163.com (J.L.)

**Keywords:** tumor immunotherapy, cell membrane coating, biomimetic nanoparticles, drug delivery

## Abstract

The advent of immunotherapy, which improves the immune system’s ability to attack and eliminate tumors, has brought new hope for tumor treatment. However, immunotherapy regimens have seen satisfactory results in only some patients. The development of nanotechnology has remarkably improved the effectiveness of tumor immunotherapy, but its application is limited by its passive immune clearance, poor biocompatibility, systemic immunotoxicity, etc. Therefore, membrane-coated biomimetic nanoparticles have been developed by functional, targeting, and biocompatible cell membrane coating technology. Membrane-coated nanoparticles have the advantages of homologous targeting, prolonged circulation, and the avoidance of immune responses, thus remarkably improving the therapeutic efficacy of tumor immunotherapy. Herein, this review explores the recent advances and future perspectives of cell membrane-coated nanoparticles for tumor immunotherapy.

## 1. Introduction

Tumor immunotherapy has become an important option in the treatment of tumors, in which tumors are treated by inducing or suppressing the immune system [1,2]. This approach is gaining popularity in research and clinical practice as a viable alternative to chemotherapy, radiotherapy, and surgery due to its unique advantages of superior safety and effective treatment results [2,3]. Over the past decade, tumor immunotherapy has evolved into many practical strategies including immune checkpoint inhibition, peripatetic cell therapy, virus lysis, and tumor vaccines [4,5]. However, the efficacy of tumor immunotherapy is still limited due to the immunosuppressive tumor microenvironment [6,7,8]. Lymphocytes play a crucial role in tumor eradication, and the immunosuppressive tumor microenvironment usually prevents the activity and infiltration of cytotoxic T lymphocytes (CTLs) [9]. Immunosuppressive cells are widely distributed in tumor sites including cells that suppress myeloid differentiation (MDSC), macrophages associated with tumors (TAM), and regulatory T cells. These cells also contribute to tumor growth by forming neointima, increasing T cells, and raising natural killer cells (NK), which leads to tumor aggressiveness, poor prognosis, and proliferation [10]. Other limitations of tumor immunotherapy also exist. For example, immune checkpoint therapy represented by PD-1 is applicable to fewer patients; CAR-T-cell immunotherapy modalities have cytokine storms and solid tumor effect defects; DNA tumor nucleic acid vaccines have poor immunogenicity and systemic immunotoxicity [11]. Therefore, the development of a completely new technical tool is necessary to improve tumor immunotherapy.

With the continuous integration of materials science technology and oncology research, the role of nanoparticles in the body’s antitumor immunity is gaining increasing attention [12]. Common nanoparticles used for tumor immunization include liposomes, mesopores, metals, etc. [13]. For example, Zhang et al. developed a targeted liposome-polycation-DNA (LPD) composite nanoparticle, and encapsulated total tumor-derived RNA in lipid nanoparticles (LNPs) to target dendritic cells (DCs). Tumor antigens are expressed and presented or cross-presented to Th and Tc cells via antigen-presenting cells (DCs) in lymph nodes, ultimately producing a specific antitumor response [14]. In addition, Mahvash Sadeghi et al. prepared Apt-modified AuNP complexes, and the gold nanoparticles caused a decrease in IgE levels and a significant increase in the expression of TGF-β and IFN-γ compared to other controls, suggesting that gold nanoparticles can improve the imunomodulatory responses of Th1 and Treg [15]. However, nanoparticles also have some disadvantages because the body’s immune system is reactive to heterogeneous substances, and immune cells in the blood and tissues tend to phagocytose and remove some nanoparticles. Monocytes, platelets, leukocytes, dendritic cells (DCs), and even resident phagocytes can take up nanoparticles in the blood and tissues, which may cause the nanoparticles to leave their intended site of action, thus reducing the efficacy of the drugs [16]. Surface modification can significantly improve the cycle time of nanoparticles, which is necessary to improve the performance of nanoparticles [17]. Thus, researchers have found cell membrane-derived nanoparticles to be a promising approach to evade immune clearance, due to the natural composition of the cell membrane [18]. Cell membrane bionanotechnology is a bionic replication of cell membrane properties that combines the natural cell membrane properties with those of an artificial inner core material, simply by separating the cell membrane from the cell and covering it with nanoparticles carrying antitumor drugs, thus greatly improving the biocompatibility of the nanoparticles [19]. Cell membrane-encapsulated nanoparticles essentially mimic the properties of cell membranes, offering advantages such as prolonged blood circulation, recognition of antigens for enhanced targeting, better cellular interactions, slow release of drugs, and reduced in vivo toxicity. Moreover, cell membrane allow the encapsulated drug to reach the target tissue and plays a role in preventing premature clearance and degradation or release of the drug [19,20] Hence, membrane bionanotechnology overcomes the limitations of surface modification of conventional nanoparticles in biomedical applications [21,22]. Most cell membranes can be used as membrane sources for biological nanoparticles [23]. Biological nanoparticles are commonly used for antigen presentation, targeted drug delivery, and immune response against tumors (Figure 1). The mechanism of action of membrane-coated nanoparticles includes targeted delivery, immune modulation, and precise therapy.

In this review, we introduce the characteristics of cell membranes and membrane coating technologies for encapsulating nanoparticles, followed by a summary of the latest representative applications of membrane-coated biomimetic nanoparticles in cancer immunotherapy, and its advantages and limitations.

## 2. Background of Membrane Coating Technology for Tumor Immunotherapy

### 2.1. Introduction of Cell Membranes

The cell membrane is a membrane structure consisting of a lipid bilayer, proteins, and polysaccharide substances at the edge of the cell [20]. The main functions of cell membranes include barrier function, material exchange, and information exchange. The cell membrane is the physical barrier between the cell and the external environment, which can prevent damage to the cell from the external environment and protect various organelles within the cell [19]. The cell membrane is also responsible for the exchange of substances between the cell and the external environment. The cell membrane is selectively permeable, so that some small molecules can pass freely, while large biological molecules (such as proteins) need to be endocytosed, to enter the cell. In addition, cell membranes contain many ion pumps that play a key role in maintaining cellular osmotic pressure and internal homeostasis. Notably, in addition to material exchange, cell membranes also mediate the exchange of information between the cell and its environment [24]. There are numerous membrane receptors on the cell membrane that play a key role in intracellular signal transduction and are essential for the regulation of cell proliferation, migration, apoptosis, metabolism, and other processes [25]. The use of different cell membranes will produce nanoparticles of different properties because of the large differences in the cell membranes.

### 2.2. Membrane Coating Technology

Membrane coating technology refers to the wrapping of biological membranes around the surface of nanoparticles. Nanoparticles coated with cell membranes have the advantages of prolonging blood circulation, recognizing antigens, enhancing targeting, better cellular interaction, slow release of drugs, and reduced in vivo toxicities [26]. There are four main approaches for membrane coating technology, namely, porous membrane coextrusion, ultrasonic processing, rapid mixing combined with electroporation of microfluidic systems, and in situ packaging based on living cells [27,28,29,30].

Porous membrane coextrusion is a method to complete membrane coating by a coexisting-extruding mixture including cell membrane vesicle preparation and vesicle nanoparticle fusion [27,31,32]. Hu’s group obtained erythrocytes by centrifugation by using hypotonic solutions to distend the erythrocytes to release their contents, and then obtained the erythrocyte membrane by centrifugation. The RBC membrane vesicles and PLGA nanoparticles are physically extruded through porous membranes, and the final mixture of red blood cell membranes is wrapped around the nanoparticle surface. This extrusion method has the advantage of being simple to operate, but has high material losses [27]. In addition to the porous membrane coextrusion method, ultrasonication is also a good choice. Ultrasonication is a method of obtaining cellular membrane carriers by means of the destructive force provided by ultrasonic energy. Ultrasonication reduces the loss of material, while the disadvantage is that ultrasonication may destroy the core of the nanoparticles and reduce the efficiency of membrane-coated nanoparticles. Spontaneous formation of core-shell nanostructures during the sonication of membrane vesicles and nanoparticles to complete the membrane coating process [28]. Fan’s group suppressed glioma growth by preparing membrane biomimetic nanoparticles using ultrasound [33]. Since the ultrasonic method may have efficiency problems, electroporation has been widely used. Furthermore, compared with the coextrusion method, electroporation avoids the repeated extrusion of nanoparticles by the porous membrane, maintains the integrity of the membrane to a certain extent, and reduces the loss of cell surface proteins, thus achieving a better treatment effect [34]. Electroporation breaks down the electrical layer on the cell membrane through an applied electric field and creates pore transients for molecules and nanoparticles to put in the microfluidic system. There are Y-shaped merging channels and S-shaped mixing channels present in the device. With finely tuned pulse voltage, timing, and flow, a membrane-coated nanoparticle is produced from electroporation [29]. Taking the erythrocyte membrane and magnetic Fe_3_O_4_ nanoparticles as an example, the two are injected into a microfluidic device. As the mixture passes through the electroporation zone, the electrical pulses effectively facilitate the entry of nanoparticles into the erythrocyte membrane vesicles. RBC membrane-coated Fe_3_O_4_ enhanced MRI and guided photothermal therapy make the treatment of breast tumors possible [29]. In addition to the above methods, in situ packaging based on living cells is also a common method for the preparation of cell membrane carriers. In this preparation scheme, the cells are co-incubated with the desired nanoparticles in a serum-free environment, cellular processes can secrete vesicles containing exogenous nanoparticles, and membrane-coated nanoparticles can be obtained by collecting the cell vesicles by centrifugation [29].

## 3. Cell Membrane-Coated Nanoparticles for Immunotherapy

Biomimetic cell membrane nanotechnology is widely used and easy to obtain, so biomimetic nanoparticles show excellent application prospects in the field of antitumor immunotherapy. Cell membranes (such as red cell membranes, leukocyte membranes, platelet membranes, cancer cell membranes, and hybrid membranes) can be used as materials for biomimetic nanotechnology. The natural functions and properties of cell membranes allow bionanoparticles to exhibit unique cell-like functions (Table 1). Researchers can choose suitable membrane materials according to different needs, based on the different properties of each type of membrane. The encapsulation of nanoparticles by erythrocyte membranes could improve the long-term cycling ability of nanoparticles and alleviate clearance by the immune system [27]. The presence of leukocyte membranes activates the immune system and targets tumors [35]. Platelet cell membrane-based bionic systems can be used to treat tumor metastasis [36]. Nanoparticles coated with the membranes of tumor cells have homologous targeting to tumors [37,38]. In contrast, the hybrid membrane retains the properties of the source cell membrane and has multiple functions [39]. In addition, further functionalization of cell membranes is possible, providing greater flexibility for nanoparticles. Here, we discuss in detail the various cell membrane-coated technologies for cancer immunotherapy.

### 3.1. Red Blood Cell Membrane-Coated Nanoparticles

Red blood cells (RBCs) are abundant in the body’s circulatory system, are widely distributed, and are easy to obtain. There are special polysaccharides and proteins on the surface of RBCs, which can significantly prolong the survival time of RBCs in the body, even reaching more than three months [30]. Therefore, coating with erythrocyte membranes can notably reduce the immunogenicity of nanoparticles and prolong their circulation time, thereby enhancing the killing effect on tumors. Based on these characteristics, red blood cell membrane coating is a promising approach for cancer therapy. There are two mechanisms by which the erythrocyte membrane coating achieves its function, namely, evading macrophage uptake and systemic clearance to extend the half-life of the drug and inducing immune responses with nanovaccines [27,54].

The CD47 molecule on the surface of the erythrocyte membrane is able to bind to signal-regulated protein alpha (SIRPα) to inhibit the corresponding signaling pathway, so that erythrocyte membrane-coated nanoparticles can avoid phagocytosis and clearance by macrophages, thus prolonging the half-life of the drug in vivo [27,55]. Protected by the red blood cell membrane, the nanoparticles can be loaded with multiple drugs to reach the tumor site smoothly, enhancing the combined therapeutic effect of the drugs. Yu’s team constructed 150-nm hyaluronidase-triggered size-reducible bionanoparticles (defined as pPP-mCAuNCs@HA), which were co-loaded with the photosensitizer dimagnesium chlorophyllide and the reactive oxygen-reactive paclitaxel (PTX) dimer prodrug (PXTK), and finally the membrane surface was further loaded with anti-PD-L1 peptide (dPPA) [40] (Figure 2). Under irradiation, ROS produced by the photosensitizer stimulates PXTK hydrolysis to release PTX, and the hydrolysis by-product cinnamaldehyde in turn stimulates the mitochondrial production of ROS, thus keeping the ROS levels high. The combined anti-PD-L1 peptide enhances the maturation of immunogenic cells (CD4 T cell, CD8 T cells, and NK cells) and the secretion of cytokines (TNF-α and IL-12) stimulated by PDT-induced immunogenic cell death. In addition, antitumor treatment was assessed in female rats bearing 4T1 tumors. The results showed that the combination of chemotherapy, PDT, and PD-L1 blockade immunotherapy increased the tumor suppression rate to 84.2%, much higher than that of PDT alone at 65.4%, showing a remarkable antitumor curative effect. This suggests that erythrocyte membrane-coated nanotechnology can synergistically inhibit tumor overgrowth and apoptosis with a variety of other therapeutic approaches.

The main mechanism of current tumor immunotherapy is to recognize and kill tumor cells by activating the patient’s own immune system. Nanotechnology-based tumor vaccines have been developed to efficiently induce specific cellular immunity by delivering antigens to antigen-presenting cells (APCs) such as dendritic cells (DCs). To enhance the stability and antigen presentation efficiency of the nanovaccine, the surface of the nanovaccine was coated with a erythrocyte membrane. Guo et al. showed that in a metastatic melanoma model, compared with other preparations, the nanovaccine (Man-RBC-NP) could effectively inhibit tumor growth and inhibit tumor metastasis, proving that erythrocyte membrane coating technology has great research potential in tumor immunotherapy [56]. Feng et al. cleverly introduced deoxyhemoglobin (Hb)-dependent drugs (RRx-001) together with Hb into TiO_2_ nanoparticles (defined as R@HTR) coated with RBC membranes to enhance the effects of acoustic dynamics therapy through multiple pathways [41]. Hb is a natural protein in red blood cells and is mainly responsible for transporting oxygen. In the hypoxic tumor microenvironment, oxygen carried by Hb enhances SDT-mediated ROS production, and deoxygenated Hb combined with RRx-001 also activates ROS and NO production, triggering ICD in tumor cells to stimulate the immune system. In addition, RNS, a product of reactive oxygen species and nitric oxide reactions, promotes a shift in TAMs to an antitumor M1-like phenotype, reducing immunosuppressive cells including regulatory T cells (Tregs) and myeloid-derived suppressor cells (MDSCs) and increasing cytotoxic T lymphocyte (CTL) infiltration. These results demonstrate that R@HTR is an effective tumor treatment strategy. Basal-like breast cancer is a type of malignant tumor with a high mutation rate and is not sensitive to traditional chemotherapy, but immunotherapy has a good effect on it. Xin Liang et al. used near-infrared (NIR) irradiation to induce breast cancer cell apoptosis and necrosis while mobilizing the immune system to eliminate cancer cells by recruiting dendritic cells (DCs). Then, they used a nanovesicle formed by coating black phosphorus quantum dots (BPQDs) with erythrocyte membranes (RM) to significantly inhibit the growth of basal-like breast cancer tumors by PTT combined with aPD-1 therapy [42].

Red blood cell membrane-coated nanoparticles can protect the antigen from immune system clearance, transfer the antigen in the organs, and directly present the antigen to the immune cells after the treatment of the antigen presenting cells, leading to sensitization and multiplication of antigen-specific CD4 and CD8 effector T cells to induce the immunity reaction to resist neoplasm [57,58,59]. In the treatment of melanoma, polymer nanoparticles encapsulated by erythrocyte membranes can be used as a nanovaccine to improve targeting and antigen presentation efficiency by inducing an immune response to fight melanoma. Guo’s team prepared PLGA nanoparticles coated with erythrocyte membranes for the delivery of antigen peptide and immune adjuvant monophosphate (MPLA), according to the principle of protein content that can be retained by the erythrocyte membrane-coated vaccine, and enhanced cell uptake in vitro [56]. By inserting mannose into the material, the nanoparticles actively target APC in the lymphoid organs, and PLGA nanoparticles are very sensitive to redox reactions. Therefore, compared with other preparations, the nanovaccine prolongs the time of tumor recurrence, shows great advantages in cancer immunotherapy, and provides a new strategy for the development of safe, effective, and low-cost bionic vaccines.

### 3.2. Leukocyte Membrane-Coated Nanoparticles

Leukocytes are important components of the human immune system. According to the different shapes and functions of cells, they can be divided into macrophages, dendritic cells, B lymphocytes, T lymphocytes, natural killer cells (NK cells), neutrophils, eosinophils, and basophils. The leukocyte membrane has specific targeting ligands and surface proteins, which have a defensive effect on the body. When inflammation occurs in the body, leukocytes undergo chemotaxis. After receiving the signal, they can quickly reach the diseased tissue from the blood vessel [60]. T lymphocytes produced by lymphoid organs are an important cellular component of the body’s immune response [61]. Based on these properties, leukocytes are also a potential membrane source for membrane-derived nanobiomimetic technology. According to the self-recognition mechanism of leukocytes, which can promote transendothelial transport to avoid lysosomes, Parodi’s group reported a leukocyte membrane-coated nanoporous silicon nanoparticle material [43]. By acting on endothelial cells, the material can change the targeting and migration properties of the cells, which can increase the nanoparticle circulation time and improve the antitumor efficacy. This shows that leukocyte membrane-derived nanoparticles are superior to ordinary nanoparticles in the treatment of cancer.

Different kinds of leukocytes can serve as membrane sources for biomimetic nanotechnology. The exact source of NK cells is not well-understood and is generally believed to derive directly from the bone marrow, where their developmental maturation is dependent on the microenvironment of the bone marrow [62]. NK-cell membrane proteins can induce macrophage M1 polarization [43]. The killing activity of NK cells is MHC restricted and antibody independent, and activated NK cells can synthesize and secrete a variety of cytokines, exerting the effect of regulating immunity and hematopoiesis and directly killing target cells. The target cells of NK cells mainly have certain tumor cells, so NK cells play a vital role in immunity for the body’s antitumor effects [62]. Deng’s group designed a photosensitizer nanoparticle material encapsulated by the NK-cell membrane based on the principle that NK-cell membrane proteins can induce macrophage M1 polarization and that inflammatory macrophage polarization can exert antitumor therapeutic effects (Figure 3). This material was able to reduce cancer cell death by enhancing PDT, thus improving the antitumor immune workpiece ratio of the NK-cell membrane. In addition, the material can selectively accumulate in the tumor, eliminate the growth of the primary tumor, and inhibit distant tumors [44,63].

T cells can also serve as a membrane source for biomimetic nanotechnology. T cells develop from pluripotent stem cells in the bone marrow and the immune response generated by T cells is regarded as cellular immunity, of which there are two main kinds: destroying the target cell membrane after specifically binding to the target cell and killing the target cell directly; the other is the release of lymphokines, which eventually allows the immune effects to expand and enhance. There are many different markers on T-cell membranes such as receptor for T-cell antigen (TCR), HLA antigens, and interleukin receptors, and these membrane surface markers are significant in inducing immune responses [64]. In response to these characteristics, Li’s group developed a new tumor microenvironment-responsive nanoparticle coated with T cells (also known as RCM@T. HA disulfide bond cloaking HA and curcumin/vitamin E succinate are incorporated into modified T-cell membranes by PBA) [65]. The membrane of T cells of this nanoparticle not only serves as a protective shell for drug delivery, but also selectively binds to the PD-L1 of tumor cells as a programmed cell death-1 (PD-1) C antibody. The cell wall of T-cell fragments targeting PD-L1 of cancerous cells for immunotherapy for cancer kill tumor cells directly and raise the level of CD8+ T cells to promote effector cytokine release to further enhance tumor killing [66]. Zhai et al. reported an epigenetic nanoinducer of T lymphocyte membrane modification endowed with programmed cell death protein 1 (PD1), which was called OPEN, to deliver the IFN inducer ORY-1001. OPEN led to a strong inhibition of xenograft tumor growth via certain sequential processes. This T lymphocyte membrane-coated epigenetic nanoinducer provided a reliable membrane biomimetic nanoparticle drug delivery platform for tumor immunotherapy [67]. Immune checkpoint blockade, as one of the mainstream tumor immunotherapies, can revive exhausted T cells and exert immunotherapy effects, but the treatment effect of most patients is not satisfactory. To solve this challenge, Kang et al. developed T cell membrane-coated nanoparticles (TCMNPs) for tumor immunotherapy and achieved promising therapeutic effects. TCMNPs can not only target tumors through T-cell membrane-initiating proteins and kill cancer cells by releasing anticancer molecules and inducing Fas ligand-mediated apoptosis, but also resist cancer by eliminating TGF-β1 and PD-L1 [68]. 

DC cells are professional antigen-presenting cells (APCs) that take up, process, and present antigens to initiate T-cell-mediated immune responses [69]. Cheng’s group designed a DC-based nanovaccine for ovarian cancer immunotherapy. The nanovaccine was prepared by encapsulating IL-2-loaded PLGA-NPs with the cell membrane of mature DCs. By presenting the functional properties of membrane proteins of DCs (e.g., MHC, CD86 and CD40) on their surface, the nanovaccine is expected to mimic the antigen presentation ability of DCs and release IL-2 in a paracrine manner, thereby activating T cells and stimulating a robust antitumor immune response without being affected by immunosuppressive tumor-associated macrophages (TAMs) and physiological barriers during migration and antigen presentation. Experimental results showed that ovarian cancer model mice that received this vaccine effectively inhibited the growth and metastasis of ovarian cancer [45]. 

Macrophages are a class of innate immune cells derived from monocytes and are mainly composed of the M1 type, which promotes inflammation, and the M2 type, which inhibits inflammation. The TAMs in the tumor microenvironment are mainly the M2 type. Studies have found that tumor proliferation, migration, recurrence, and other processes are closely related to TAMs. TAMs can specifically recognize tumor cells and exosomes [69], so biomimetic nanoparticles constructed by macrophage membrane coating technology are an effective tumor targeting method. Zhang et al. achieved the targeting of breast carcinoma in situ and lung metastases by coating a versatile ROS-cleavable camptothecin (CPT) prodrug (DCC) with a TAM membrane, and achieved the targeting of breast carcinoma in situ and lung metastasis by ROS-triggered CA/CPT release and CA/CPT- mediated ROS generation induces tumor ICD [70]. Notably, this study also found that macrophage membrane-coated DCC could interfere with the interaction of TAMs with tumor cells, enhance tumor-specific T cell infiltration, and activate anti-tumor immune responses. In addition, Wang et al. used the primary macrophage membrane to coat gold nanocages carrying the PD-L1 antibody and TGF-β inhibitor and realized the synergistic effect of PTT and combined immunotherapy on colon cancer cells [71]. In conclusion, macrophage membrane coating technology can improve the targeting of nanoparticles and is an effective means of regulating the tumor microenvironment and combining antitumor therapy.

### 3.3. Platelet Membrane-Coated Nanoparticles

Platelets are a group of disc-shaped, nonnucleated cell fragments approximately 2–4 µm in diameter produced by mature megakaryocytes in the bone marrow, and their main functions are hemostasis and thrombosis [72]. Platelets also play a key role in cancer, atherosclerosis, and bacterial infections due to their ability to promote inflammation and participate in immune responses [73]. In the process of cancer cell metastasis, platelets can participate in the regulation of tumor cell epithelial–mesenchymal transition (EMT) by releasing extracellular vesicles [74]. In addition, platelets can also target tumor cells by utilizing special receptors and glycoprotein distributions on the surface of tumor cell membranes such as podoplanin, PSGL-1, disintegrin, and metalloproteinase domain-containing protein 9 (ADAM-9), and fibrinogen/αvβ3, which play an antitumor effect [72]. Baharak et al. used the platelet membrane-coated TLR7 agonist R848 to construct an antitumor nanoparticle (PNP-R848) capable of activating innate immunity [46]. The authors found that the nanoparticles could target the colon cancer and breast cancer in tumor-bearing models, using R848 stimulation to create an inflammatory microenvironment at the tumor site, and activate DCs and T cells, thereby inhibiting tumor growth and metastasis. In addition, Chen et al. achieved chemotherapy combined with immunotherapy for acute myeloid leukemia by using a platelet membrane to wrap the tumor chemotherapy drug doxorubicin (DOX) and the natural antitumor drug ginsenoside (Rg3) [47]. In summary, platelet membrane coating technology is an effective strategy for regulating the tumor immune microenvironment and can be used in combination therapy with immunotherapy.

### 3.4. Tumor Cell Membrane-Coated Nanoparticles

Tumor cells can be cultured in vitro with unlimited value-added properties. In tumor immunotherapy, the cell membrane of cancer cells can also serve as a membrane source for membrane-coated biomimetic nanotechnology [75]. There are many aspects that make tumor cells unique, and their membrane coatings can escape the immune system and target homologous cells, allowing them to extend their cell cycle. Biomimetic tumor nanovaccines, which can induce multiple tumor antigen immune responses, have been widely studied in the field of tumor immunotherapy in recent years [76]. Its advantage is that nanoparticles designed by employing biomimetic nanotechnology retain tumor-specific antigens and can improve the homologous targeting ability for the smooth delivery of tumor drug targeting, which enriches the strategy for tumor immunotherapy. Therefore, the cell membranes of cancer cells deserve to be explored as a membrane source for membrane-coated nanobiomimetic technologies [77]. Luo et al. used the cell membrane of the breast cancer cell line 4T1 to coat triphenylphosphine (TPP) metal zirconium framework material and added the TLR agonist R837 [Zr-TCPP(TPP)/R837@M] [48]. The material can target mitochondria after being taken up by tumor cells and can induce the immunogenic death of tumor cells under the action of ultrasound. At the same time, the released R837 can promote the maturation of DC cells and enhance their antigen-presenting ability. Furthermore, the nanoparticle was able to synergize with the anti-CTLA4 antibody in vivo to reverse the immunosuppressive microenvironment. In vivo experiments have shown that the nanoparticles can effectively inhibit tumor recurrence and inhibit tumor metastasis. Wang et al. constructed a poly (d,l-lactide-glycolide) (PLGA)-based intelligent bionic nanoplatform (AFT/2-BP@PLGA@ MD) [50]. The nanoparticles used the PLGA polymer to encapsulate 2-bromo-palmitate (2-BP) and the antitumor drug afatinib (AFT), and used tumor cell membranes to encapsulate PLGA nanoparticles to enhance their homologous targeting and immune escape ability. At the same time, the study also immobilized the matrix metalloproteinase-2 (MMP-2) substrate polypeptide-linked PD-L1 inhibitor D-peptide antagonist (DPPA-1) on the outer membrane through fatty acid molecules to achieve DPPA-1 in the tumor. The nanoparticles can target tumors in vivo, inhibit tumor cell proliferation, promote tumor cell apoptosis, block PD1/PD-L1 signal binding between tumors and T cells, inhibit tumor immune escape, and promote T cell activation. Studies have shown that the nanoparticles can effectively inhibit the metastasis and recurrence of triple-negative breast cancer. Liu’s team also constructed a tumor vaccine. They loaded Toll-like receptor agonists into PLGA nanoparticles as the core, wrapped the tumor cell membrane on the outer layer, and later modified the surface with mannose modification to construct a cancer vaccine (Figure 4). The surface proteins on the cell membrane can be taken up by antigen-presenting cells (e.g., dendritic cells), and mannose specifically binds to dendritic cells (DCs), enhancing the stimulation of DCs and triggering DC maturation. The ultimate goal is to retard tumor progression and improve the efficacy of melanoma tumor treatment [78]. Similarly, the Meng group designed an H460 lung cancer cell membrane-coated superparamagnetic iron oxide nanoparticle and conjugated PD-L1 inhibitory peptide (TPP1) and MMP2 substrate peptide (PLGLLG) to the H460 membrane. The purpose of inhibiting tumor growth was achieved by homotypic targeting of the H460 cell membrane and specific digestion by the MMP2 enzyme, delivering and releasing the TPP-1 peptide that inhibited PD-L1 and PD-1 binding into the tumor microenvironment [79].

### 3.5. Hybrid Membrane-Coated Nanoparticles

Although a single membrane coating technology can provide tumor targeting and immune evasion capabilities for loaded nanoparticles, multifunctional cell membrane coating technology will provide a more convenient and flexible design scheme for nanoparticle delivery. Hybrid membrane coating technology is a biomimetic membrane coating technology that fuses two or more cell membranes to encapsulate nanoparticles. The hybrid membrane-coated nanoparticle mainly includes the three steps of cell membrane extraction, cell membrane fusion, and nanoparticle encapsulation [80]. During cell membrane extraction, the cell membrane can be obtained by swelling with hypotonic solution for red blood cells and platelets without nuclei [81]. For eukaryotic cells, cell membranes can be obtained by repeated freezing and thawing, ultrasonic damage, and hypotonic solution swelling [77]. The obtained cell membranes from different sources can be mixed by stirring, ice baths, or ultrasonic treatment to obtain hybrid membranes [39]. The hybrid membrane coated nanoparticles can be produced by physical extrusion, ultrasound, or microfluidic-based methods [82]. Nanoparticles coated with hybrid membrane technology will present more abundant properties. Lin et al. fused macrophage membranes with a thylakoid (TK) membrane to coat hollow mesoporous Prussian blue (HMPB) nanoparticles with mannose decoration and hydroxychloroquine (HCQ) adsorption (TK-M@Man-HMPB/HCQ) [50]. Due to the coating of the hybrid membrane, the nanoparticles were able to effectively avoid phagocytosis, enhance tumor site accumulation, and circumvent the hypoxic conditions of the tumor microenvironment. At the same time, with the degradation of HMPB, the nanoparticles could induce M2-type macrophages to differentiate into M1-type macrophages by releasing iron ions and HCQ, and cytotoxic T cells (CTLs) can kill tumors [50]. Zhao et al. constructed a nanoparticle by coating the immune adjuvant R837 with a hybrid membrane derived from tumor cells and DC cells [51]. The nanoparticles could target DCs in the tumor microenvironment, enhance their tumor cell antigen presentation ability, and exert anti-tumor effects by activating both CD4+ and CD8+ T cells. In addition to directly regulating the tumor immune microenvironment, hybrid membrane technology is often used in the study of combined anti-tumor therapy such as chemoimmunotherapy and photothermal immunotherapy. Zang et al. used hybrid membrane technology to coat tumor cell membranes and activated fibroblast membranes with a glycolysis inhibitor (PFK15) and the chemotherapeutic drug paclitaxel (PTX) to form a dual-targeted nanodrug delivery system [52]. The system can simultaneously target tumor cells and tumor-associated fibroblasts (CAFs) and block the glycolytic pathway of both. When the glycolytic metabolic pathway is inhibited, the production of lactic acid is downregulated, thereby ameliorating the killing effect of the immune system on tumors. At the same time, when the glycolytic pathway of tumor cells is inhibited, their sensitivity to chemotherapeutic drugs is improved, and the inhibitory effect of PTX carried by this system on tumors is improved. In addition, due to the dual targeting of the hybrid membrane, the system can further block the metabolic support effect of CAFs on tumor cells, so it can exert further antitumor effects. Xiong et al. fused mouse-derived ID8 ovarian cancer cell membranes with red blood cell (RBC) membranes to form a hybrid biomimetic coating (IRM) and encapsulated indocyanine green (ICG)-loaded magnetic nanoparticles (Fe_3_O_4_-ICG@IRM) for combination therapy in ovarian cancer [53]. Fe_3_O_4_-ICG@IRM retains both ovarian cancer and RBC cell membrane proteins, is highly specific for ovarian cancer cells, and significantly prolongs the circulation time of nanoparticles in blood. Interestingly, in a bilateral tumor model, the hybrid membrane-coated nanoparticles were able to activate specific immunity and kill syngeneic ID8 tumor cells. In addition, Fe_3_O_4_-ICG@IRM nanoparticles are photothermally responsive and can increase the temperature by near-infrared light irradiation, induce tumor cell necrosis and release tumor antigens, further activate CD8+ cytotoxic T cells, and inhibit regulatory Foxp3+ T cell differentiation, thereby enhancing antitumor immunotherapy for primary and metastatic tumors. Therefore, hybrid membrane coating technology further expands the source of cell membranes and provides convenience for the design of antitumor biomimetic nanoparticles with potent antitumor activity.

## 4. Conclusions and Outlook

This review mainly introduces the application of membrane-coated nanoparticles from different cell membranes in tumor immunotherapy. Membrane bionanotechnology enriches nanoparticle functions by wrapping cell membranes around the surface of the nanoparticles. These methods not only overcome the rejection reaction of the body but also endow the membrane-coated biomimetic nanoparticles with good biocompatibility, which can improve the circulation time of nanoparticles in the body and prolong the half-life of nanoparticles. In addition, biomimetic nanotechnology of the cell membrane based on tumor cells can also enhance the targeting of drug delivery. Therefore, biomimetic nanotechnology is a promising new method to address the current dilemma of antitumor immunotherapy, but it still needs to overcome many challenges to make it widely used in clinical practice. For example, the preparation method of membrane biomimetic nanotechnology is complex, the efficiency is low, the synthesis scale is small, the preservation is difficult, and the stability of the physical and chemical fusion cell membrane on the material surface limits the promotion of this technology. This also points out the direction for future research such as simplifying the preparation method, expanding and scaling up, and improving the stability of the fusion membrane on the cell surface. In the future, this new cell membrane coating technology can provide great prospects in drug targeted delivery, biological detoxification, immune regulation, and phototherapy. For example, ligands composed of antibodies, peptides, and proteins can be integrated into the cell membrane to improve the function of cancer immunotherapy. It is also possible to prepare stimulants to the surface of the cell membrane by simple methods to develop responsive nanoparticles such as photoreponsive nanoparticles. Cell membrane coating technology can also be combined with other biomaterials such as hydrogels or microspheres to improve their antitumor activity. The artificial membranes composed of natural or synthetic polymers such as chitosan or dextran could provide an alternative to cell membranes, due to their large-scale production potential, safety, and biocompatibility. However, it is still difficult for these artificial membranes to completely acquire the full functions of cell membranes due to the lack of many critical membrane protein, saccharides, and lipids in the artificial membranes. Therefore, membrane-coated biomimetic nanoparticles represent a state-of-the-art multifunctional weapon for tumor immunotherapy.

## Figures and Tables

**Figure 1 membranes-12-00738-f001:**
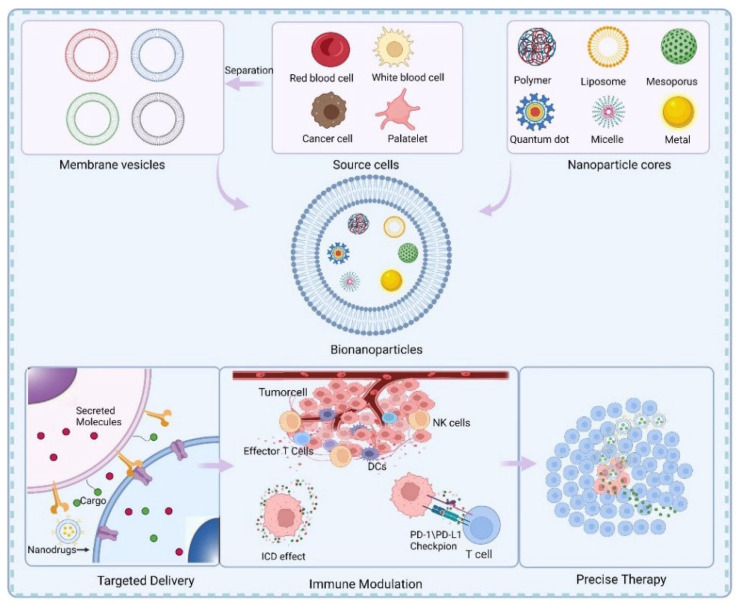
A schematic diagram of the preparation of membrane-coated biomimetic nanoparticles and their functions. The biocompatibility and functionality of nanoparticles could be improved by means of membrane coating. A wide range of cell sources can be used for membrane sources including blood red blood cells, white blood cells, tumor cells, platelets, etc. The membrane-coated biomimetic nanoparticles could effectively regulate the immune microenvironment through various effects such as immunogenic cell death (ICD) induction, ultimately achieving the precise treatment of tumors.

**Figure 2 membranes-12-00738-f002:**
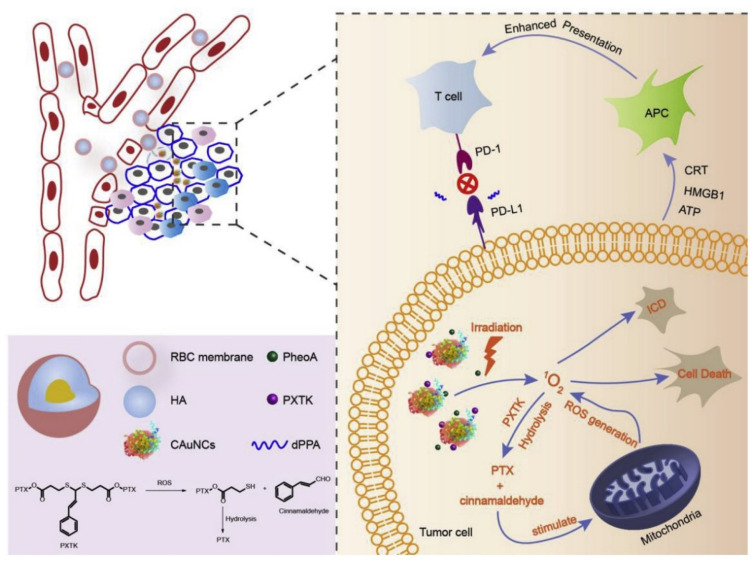
A schematic diagram of enhanced cancer immunotherapy in response to hyaluronidase gold nanoparticles combined with PDT. In the presence of photosensitizers and irradiation, ambient oxygen can be converted into ROS such as singlet oxygen (^1^O_2_), leading to apoptosis and necrosis. In addition, ROS generated by PDT induces the release of PTX from the precursor drug PXTK to kill tumor cells, while PD-L1 alleviates immunosuppression and enhances the host immune system. Reproduced with permission from ref. [40] copyright 2019, Biomaterials.

**Figure 3 membranes-12-00738-f003:**
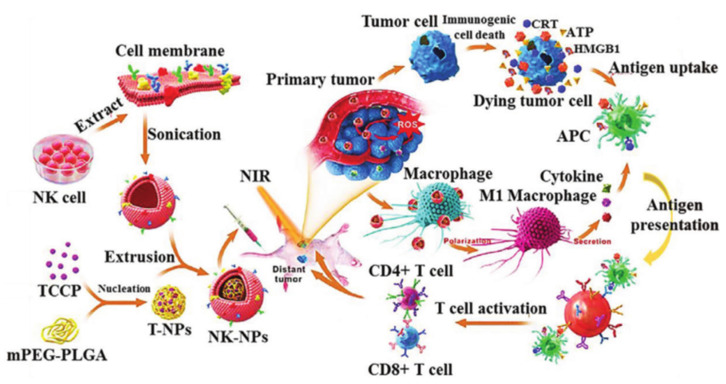
A schematic of the NK-cell membrane-based nanoparticles for PDT-enhanced cancer immunotherapy. Extracted NK-cell membranes were coated by extrusion onto polymeric nanoparticles loaded with the photosensitizer TCPP. The NK-cell membrane enables NK NPs to polarize M1 macrophages in tumors. NK NPs can induce the production of damps from dying tumor cells via PDT-induced immunogenic cell death (ICD) to enhance NK-cell membrane based immunotherapy. The therapy can also significantly improve the infiltration of cytotoxic T cells in tumors, resulting in highly efficient suppression of both primary and distant tumors. Reproduced with permission from ref. [44] copyright 2018 American Chemical Society.

**Figure 4 membranes-12-00738-f004:**
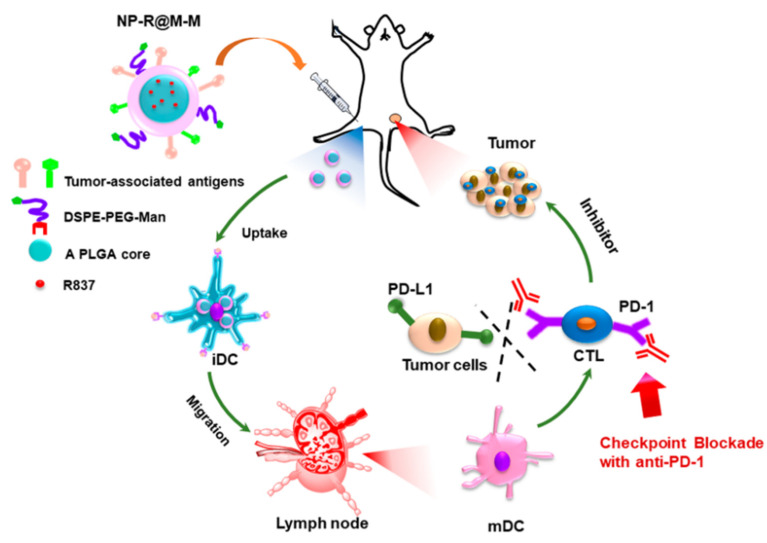
A schematic illustration of the induction of antitumor immune function by the NV based on the melanoma cell membrane as a tumor-specific antigen. Mannose specifically binds to its receptor on dendritic cells (DCs), prompting DC maturation, which works in concert with immune checkpoint inhibition on tumor cells. Reproduced with permission from ref. [78] copyright 2018 American Chemical Society.

**Table 1 membranes-12-00738-t001:** A summary of the cell membrane-coated nanoparticles.

Types of Cell Membrane	Core Nanoparticles	Applications	References
RBCM	Hyaluronidase-responsive nanoparticles	Metastasis, PDT, cancer immunotherapy	[40]
	Hollow mesoporous TiO_2_	cancer immunotherapy	[41]
	Black phosphorus quantum dot	PTT, tumor immunotherapy	[42]
WBCM	Silicon nanoparticle	PDT, cancer immunotherapy	[43]
	Hyaluronidase responsive nanoparticles	Cancer immune chemotherapy combination	[44]
	PLGA nanoparticle	Cancer immunotherapy	[45]
Platelet membrane	Polylactic acid (PLA) nanoparticle cores	Cancer immunotherapy	[46]
	DOX/Rg3 liposomes	Cancer immunotherapy	[47]
Cancer cell membrane	Triphenylphosphine (TPP) metal zirconium framework	Cancer immunotherapy, tumor metastasis	[48]
	(PLGA)-based intelligent bionic nanoplatform	Cancer immunotherapy	[49]
Hybrid membrane	A hollow mesoporous Prussian blue (HMPB) nanoparticles	Cancer immunotherapy	[50]
	Mesoporous silica nanoparticle	cancer immunotherapy	[51]
	Solid lipid nanoparticles	cancer immunotherapy	[52]
	(ICG)-loaded magnetic nanoparticles	Cancer immunotherapy	[53]

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
