# Peer review of "Membrane-Coated Biomimetic Nanoparticles: A State-of-the-Art Multifunctional Weapon for Tumor Immunotherapy"

_membranes, 2022, doi:10.3390/membranes12080738_

Round 1

Reviewer 1 Report

Overall, this is a very nice review which would benefit from addressing the points below.

Figure 1 is attractive but it is not clear to me what it is trying to show.  Maybe an extended legend would help here?

In section 2.1, some of the wording is unusual and could do with checking.

I think the start of 3.1. could be more clearly written. I’m not sure about the relevance of hemoglobin here?

Overall, figures probably need more explanation.

Some of the formatting of subsection headings is inconsistent e.g. 3.1.

3.5. – would like more detail on how the hybrid membranes are made.

It would be helpful to explicitly talk (briefly) about nanoparticles coated in artificial membranes as, although beyond the scope of the review, the topics are highly related.

Author Response

  1. Figure 1 is attractive, but it is not clear to me what it is trying to show. Maybe an extended legend would help here?

Answer: Many thanks for your kind comments. We briefly introduce the classes of membrane-coated nanoparticles, the origin of the membranes, the construction process and the mechanism of action in the figure legend of Figure1, which will explain the membrane-coated biomimetic nanotechnology more clearly.

  1. In section 2.1, some of the wording is unusual and could do with checking.

Answer: Many thanks for your kind comments. We have rewritten section 2.1 to introduce the structure, composition and function of cell membranes in more detail.

  1. I think the start of 3.1. could be more clearly written. I’m not sure about the relevance of hemoglobin here?

Answer: Many thanks for your kind comments. We made an oversight in the previous manuscript, and the reference to hemoglobin at the beginning of section 3.1 should be changed into red blood cell. We have corrected that.

  1. Some of the formatting of subsection headings is inconsistent e.g. 3.1.

Answer: Many thanks for your kind comments. We double-checked the manuscript and have changed the formatting of some subsection headings to keep them consistent.

  1. Would like more detail on how the hybrid membranes are made.

Answer: Many thanks for your kind comments. We have inserted the introduction of hybrid membrane construction techniques and commonly used encapsulation nanoparticle techniques in section 3.5. These will be helpful for the authors to have a deeper understanding of hybrid membrane encapsulation techniques.

  1. It would be helpful to explicitly talk (briefly) about nanoparticles coated in artificial membranes as, although beyond the scope of the review, the topics are highly related.

Answer: Many thanks for your kind comments. We have checked explicitly talk about nanoparticles coated in artificial membranes in the Discussion of the revised manuscript

Reviewer 2 Report

The present manuscript describes the main techniques for membrane-coating of nanoparticles to evade the immunological system and thus prolonging the circulation time of the nanodevices.  Furthermore, it reviews some of the latest achievement in the field.

The manuscript is very well written and convey suitably the main aspects of this technique. Proof of the relevance of the topic is the publication very recently of a similar review: https://doi.org/10.1016/j.cclet.2021.10.057

Minor typos: Line 283, Line 287, Line 301, Line 314

Hence, I recommend the publication of this review basically as is.

Author Response

  1. Minor typos: Line 283, Line 287, Line 301, Line 314

Answer: Thank you very much for your review. Several errors you mentioned have been corrected in our revised version

Reviewer 3 Report

The manuscript by Zhang et al, presents a compilation of works focused on the use of cell membranes as biomimetic solutions for nanoparticle stabilization and delivery. As such it is an interesting and timely effort. But as it stands now it needs more work to reach a level at which it can be published. More concrete issues follow:

- 'Common nanomaterials used for tumor immunization include liposomes, nanogels, metals, dendritic molecular nanovesicles'. The last example, dendritic molecular nanovesicles, is very specific and not common at all 

- 'Due to their nanoscale particle size, nanoparticles evade immune clearance'. It is well known that NP accumulate in 'filtering' organs where mainly macrophage eliminate them from circulation. So this statement is not very accurate. The authors themselves mention this point later in the text.

- 'There are five main approaches for membrane coating technology, namely, porous membrane coextrusion, ultrasonic processing, rapid mixing combined with electroporation of microfluidic systems, and in situ packaging based on living cells'. Those are only 4 approaches, not 5

- ref 26 is not immunotherapy. It is mentioned in a general section, but even like that an immuno related ref would be better suited there.

- In the beginning of section 3 (and in general along the whole document), there is a confusion on which material encapsulates which. It is the membranes that encapsulate the nanomaterials, not the other way around.

- In section 3.1, what is the role of hemoglobin in this application? Why is it mentioned there?

- line 154, CD47 can enhance circulation, but how do RBC membranes contribute to targeting?

- The description of ref 40 in section 3.1 line 160 is very bad. PXTK is not an antiPD-L1 peptide but  paclitaxel derivative... Re-write

- Ref 41 is not about immunotherapy but PDT chemo

- The same with ref 42. It is PTT not immuno

- Description of ref 55 is very poor

- ref 56 as described does not include immunotherapy. same with ref 57

- ref 59 very poorly presented. It starts by saying that DC membranes were used, but then says that it was tumor cell membranes...

- In ref 60 it is not clear if they used only DC membranes ir DC plus tumor cell membranes, in which case it would be described in the wrong section

- In ref 65 as described in here, the immunotherapy is too weak coming only from the fact that they use macrophage membranes

- in ref 62 there is no immunotherapy

- there is no immunotherapy in ref 67

- overall the platelet section is very poor without convincing examples on immunotherapy

- no immunotherapy in ref 73. the same in 74

- the mixed nature of the NPs in ref 78 is clear. The immunotherapy aspect is not

- ref 79, no immunotherapy. Ref 80, immuno not clear.

Overall, the subject is interesting but the focus on immunotherapy is lacking. Most of the examples do use membranes, but they don't focus on immunotherapy. The selected references used as examples need to be better chosen.

Author Response

  1. 'Common nanomaterials used for tumor immunization include liposomes, nanogels, metals, dendritic molecular nanovesicles'. The last example, dendritic molecular nanovesicles, is very specific and not common at all.

Answer: Many thanks for your kind comments. We carefully checked the manuscript, and we agree with you that dendritic molecular nanovesicles are not commonly used in the field of tumor immunotherapy, and we have removed the relevant content in the text.

  1. 'Due to their nanoscale particle size, nanoparticles evade immune clearance'. It is well known that NP accumulate in 'filtering' organs where mainly macrophage eliminate them from circulation. So this statement is not very accurate. The authors themselves mention this point later in the text.

Answer: Many thanks for your kind comments. We highly agree with you that nanoparticles can be phagocytosed and cleared by macrophages, so we have amended the statement in the manuscript.

  1. 'There are five main approaches for membrane coating technology, namely, porous membrane coextrusion, ultrasonic processing, rapid mixing combined with electroporation of microfluidic systems, and in situ packaging based on living cells'. Those are only 4 approaches, not 5

Answer: Many thanks for your kind comments. We apologize for the problem of the types of membrane coating technology, which should be 4.

  1. Ref 26 is not immunotherapy. It is mentioned in a general section, but even like that an immuno related ref would be better suited there.

Answer: Many thanks for your kind comments. We have changed that reference.

  1. In the beginning of section 3 (and in general along the whole document), there is a confusion on which material encapsulates which. It is the membranes that encapsulate the nanomaterials, not the other way around.

Answer: Many thanks for your kind comments. We have carefully corrected the misleading sections to ensure that readers will correctly understand that the nanoparticles are coated with cell membranes and not the other way around.

  1. In section 3.1, what is the role of hemoglobin in this application? Why is it mentioned there?

Answer: Many thanks for your kind comments. We carefully checked the manuscript and the reference to hemoglobin at the beginning of section 3.1 was translated incorrectly and should be red blood cell.

  1. Line 154, CD47 can enhance circulation, but how do RBC membranes contribute to targeting?

Answer: Many thanks for your kind comments. The erythrocyte membrane encapsulated material in line 154 of the original manuscript can enhance the anti-tumor effect by prolonging the blood retention time through CD47, thus increasing the duration of action of the piggybacked nanoparticles with the tumor, instead of targeting.

  1. The description of ref 40 in section 3.1 line 160 is very bad. PXTK is not an antiPD-L1 peptide but paclitaxel derivative... Re-write

Answer: Many thanks for your kind comments. We carefully checked the manuscript and modified this part.

  1. Ref 41 is not about immunotherapy but PDT chemo

Answer: Many thanks for your kind comments. We carefully checked the manuscript and modified this part.

  1. The same with ref 42. It is PTT not immune

Answer: Many thanks for your kind comments. We carefully checked the manuscript and modified this part.

  1. Description of ref 55 is very poor

Answer: Many thanks for your kind comments. We carefully checked the manuscript and modified this part.

  1. Ref 56 as described does not include immunotherapy. Same with ref 57

Answer: Many thanks for your kind comments. We carefully checked the manuscript and modified this part.

  1. Ref 59 very poorly presented. It starts by saying that DC membranes were used, but then says that it was tumor cell membranes...

Answer: Many thanks for your kind comments. We carefully checked the manuscript and modified this part.

  1. In ref 60 it is not clear if they used only DC membranes ir DC plus tumor cell membranes, in which case it would be described in the wrong section

Answer: Many thanks for your kind comments. We checked ref 60 and found that he really does not fit in this section, so we removed it. We apologize for that mistake.

  1. In ref 65 as described in here, the immunotherapy is too weak coming only from the fact that they use macrophage membranes

Answer: Many thanks for your kind comments. We carefully checked the manuscript and have rewritten this part.

  1. In ref 62 there is no immunotherapy

Answer: Many thanks for your kind comments. We carefully checked the manuscript and have rewritten this part.

  1. There is no immunotherapy in ref 67

Answer: Many thanks for your kind comments. We carefully checked the manuscript and have rewritten this part.

  1. Overall the platelet section is very poor without convincing examples on immunotherapy

Answer: Many thanks for your kind comments. We carefully checked the manuscript and have rewritten this part.

  1. No immunotherapy in ref 73. The same in 74

Answer: Many thanks for your kind comments. We carefully checked the manuscript and have rewritten this part.

  1. The mixed nature of the NPs in ref 78 is clear. The immunotherapy aspect is not

Answer: Many thanks for your kind comments. We carefully checked the manuscript and have rewritten this part.

  1. Ref 79, no immunotherapy. Ref 80, immuno not clear.

Answer: Many thanks for your kind comments. We carefully checked the manuscript and have rewritten this part.

Round 2

Reviewer 3 Report

In the replies to the reviewer concerns, the authors ackniwledged some problems and claimed to have extensively changed the manuscript. However, the manuscript is still the same. No references or the text itself have been changed.

Round 3

Reviewer 3 Report

It is clear that the authors have made a great effort to improve the manuscript that now reads much better and its content is much more in accordance to the title. There are small things like line 52-53 : 'Common nanoparticles used for tumor immunization include liposomes, mesopores, metals,' where mesopores are not nanoparticles, or refs 63 and 83 where the link with immuno is a bit weak (only from the fact the effect of the nanostructures is quoted as ICD), but in general the quality of the manuscript has really increased and in my opinion is now ready for publication.